# Antifungal and Antibiofilm Activity of Colombian Essential Oils against Different *Candida* Strains

**DOI:** 10.3390/antibiotics12040668

**Published:** 2023-03-29

**Authors:** Jennifer Ruiz-Duran, Rodrigo Torres, Elena E. Stashenko, Claudia Ortiz

**Affiliations:** 1Escuela de Microbiología y Bioanálisis, Universidad Industrial de Santander, Bucaramanga 680002, Colombia; jennifer.ruiz@correo.uis.edu.co; 2Grupo de Investigación en Bioquímica y Microbiología, Bucaramanga 680002, Colombia; rodrigotorres4@hotmail.com; 3Centro de Cromatografía y Espectrometría de Masas, CROM-MASS, Escuela de Química, Universidad Industrial de Santander, Bucaramanga 680002, Colombia; elena@tucan.uis.edu.co

**Keywords:** essential oils, antifungal agents, antibiofilm, *Candida* spp., antimicrobial activity, hyphae

## Abstract

Most *Candida* species are opportunistic pathogens with the ability to form biofilms, which increases their resistance to antifungal drug therapies and the host immune response. Essential oils (EOs) are an alternative for developing new antimicrobial drugs, due to their broad effect on cellular viability, cell communication, and metabolism. In this work, we evaluated the antifungal and antibiofilm potential of fifty EOs on *C. albicans* ATCC 10231, *C. parapsilosis* ATCC 22019, and *Candida auris* CDC B11903. The EOs’ antifungal activity was measured by means of a broth microdilution technique to determine the minimum inhibitory and fungicidal concentrations (MICs/MFCs) against the different *Candida* spp. strains. The effects on biofilm formation were determined by a crystal violet assay using 96-well round-bottom microplates incubated for 48 h at 35 °C. The EOs from *Lippia alba* (Verbenaceae family) carvone-limonene chemotype and *L. origanoides* exhibited the highest antifungal activity against *C. auris*. The *L. origanoides* EOs also presented antifungal and antibiofilm activity against all three *Candida* spp., thus representing a promising alternative for developing new antifungal products focused on yeast infections, especially those related to biofilm formation, virulence factors, and antimicrobial resistance.

## 1. Introduction

*Candida* spp. are opportunistic pathogens that pose a major threat to immunosuppressed and susceptible patients causing systemic infections with high mortality rates. The most common cause of severe and superficial fungal infections is *C. albicans*, followed by *C. auris*, *C. glabrata,* and *C*. *parapsilosis* [1]. In recent years, antifungal resistance has become an increasing problem, with approximately 7% of all *Candida* spp. bloodstream infections being resistant to drugs used for medical treatment [2,3]. Recently, the WHO published the fungal priority pathogen list to guide research, development, and public health action. This list includes *C. auris*; *C. albicans* as critical priority fungi; and *C. glabrata*, *C. tropicalis*, and *C. parapsilosis* as high priority due to their pathogenicity and drug resistance [4].

*Candida* strains are able to adhere to mucosal and synthetic surfaces forming highly complex structures called biofilms. Biofilms represent an important virulence factor that increases mortality rates, antifungal therapy costs, and length of hospital stays in affected patients [5,6]. These biofilms display increased resistance to environmental factors, the host immune response, and currently available antifungal drug classes. Some studies suggest that *Candida* spp. can alternate between two modes of cell growth, planktonic and sessile lifestyles, allowing them to thrive in multiple environments avoiding conventional fungal therapies and host defenses [7]. Currently, conventional antifungal drugs have lost efficiency against *Candida* biofilms and have shown some limitations related to their high toxicity and costs. Another virulence factor related to biofilm formation and cell adhesion is the morphological transition in *C. albicans* to hyphal structures. The hyphal form of *C. albicans* is resistant to phagocytosis and facilitates colonization, adhesion, and cell penetration [8,9]. Therefore, hyphal formation is essential for invasion, immune evasion, and biofilm formation, making it an important target for new antimicrobial compounds.

Thus, there has been an increasing interest in the search for new antimicrobial compounds with antibiofilm activity. Among these, natural compounds (e.g., essential oils, EOs) derived from different plants and their secondary metabolites have been identified as promising bioactive molecules because of their biological activities ranging from anti-fungal activity to antivirulence properties [10,11,12]. Plant-derived EOs are a mixture of large and diverse secondary metabolites obtained from plants with interesting biological properties [13,14,15]. For this reason, EOs from tropical and Colombian plants have been evaluated for their antimicrobial activity against planktonic and sessile microorganisms [11,16,17]. Previous studies have demonstrated that some EOs could exhibit several antimicrobial activities, such as antibiofilm, antifungal, and antibacterial activities [18,19,20,21,22]. In this work, we evaluated the antimicrobial potential of 50 EOs from Colombian plants against *C. albicans* ATCC 10231, *C. parapsilosis* ATCC 22019, and *Candida auris* CDC B11903 to identify promising plants and their secondary metabolic compounds with both potential antibiofilm and antifungal activities.

## 2. Results

### 2.1. Essential Oils

Fifty EOs were obtained from different plants and chemically characterized by GC/MS (Appendix A). Information on the plants and chemical composition of the EOs with high biological (antifungal or antibiofilm) activity against *Candida* strains are shown in Table 1.

### 2.2. Determination of Antifungal Activity

To identify EOs with antifungal activity, 50 EOs were initially screened against planktonic *C. albicans* ATCC 10231, *C. auris* CDC B11903, and *C. parapsilosis* ATCC 22019 at a final concentration of 750 µg/mL (Figure 1). The EOs with inhibition over 50% were considered for further testing to determine the MIC_50_ values. Antifungal activities were highly correlated to plant species and chemical compositions. For EOs with antifungal activity, the MIC and MFC values were also determined (Table 2). Thirteen EOs showed relevant activity against *Candida* strains. The highest antifungal activity was presented by the EO LACL from *Lippia alba*, carvone-limonene chemotype against *C. auris*. Additionally, the EOs derived from *L. origanoides* plants (LOC, LOCpT, LOT-I, LOTC, and LOT-II) showed inhibitory activity against all of the studied strains. Spearman’s correlation (r^2^) between the EOs’ constituents and the antifungal activity (Appendix A) indicates that compounds like thymol, thymyl methyl ether, carvacrol, γ-terpinene, and *p*-cymene have a negative correlation, which means that a higher concentration of the compound would probably result in lower MICs/MBICs, thus better antimicrobial activity.

### 2.3. Effect on Biofilm Formation

Initially, all of the EOs were evaluated at sub-inhibitory concentrations corresponding to their MIC_50_ values (Appendix A). Results indicated that subinhibitory concentrations of *L. origanoides* EOs (90–188 µg/mL) diminished fungal adhesion in all three strains evaluated. Minimum biofilm inhibitory concentration (MBIC_50_) values are presented in Table 3 and Appendix A. It is important to highlight that EOs with antibiofilm activity were predominantly derived from *L. origanoides* plants (Figure 2; Table 3).

Spearman’s correlation (r^2^) between the EOs’ constituents and antibiofilm activity (Appendix A) showed that thymol, thymyl methyl ether, carvacrol, and γ-terpinene presented a higher correlation to the inhibition of biofilm formation, in addition to their antifungal potential in planktonic cultures. Additionally, *trans-*β-bergamotene, *cis*-β-ocimene, humulene epoxide II, and terpinen-4-ol were mainly related to antibiofilm activity, while iso-menthone was correlated to antibiofilm activity exclusively for *C. auris*.

### 2.4. Scanning Electron Microscope (SEM) Analysis

The SEM analysis allowed us to visualize biofilm formation in *C. auris* after growing for 24 h with and without EO treatments (Figure 3). Subinhibitory concentrations of EO LOT-I (Figure 3B) and LOT-II (Figure 3C) showed significant decreases in cellular adhesion to the glass surface. These results agreed with the inhibition biofilm assays shown above (Figure 2).

### 2.5. Inhibition of Hyphal Formation in C. albicans

The microscopic observation of *C. albicans* treated with bioactive EOs showed inhibition hyphal formation at subinhibitory concentrations for the *L. origanoides* EOs (LOT-I, LOTC, and LOT-II) and *Minthostachys mollis* (MM-II) (Figure 4). The EOs from *L. origanoides* inhibited hyphal formation in *C. albicans* compared to the control group, evidenced by the marked decrease of yeast cells with hyphal morphology. The EO LACL presented a slight inhibition of hyphal formation because only a decrease in hyphal length was observed, but not in the number of hyphae in yeast cells. Cells with hyphae and pseudo hyphae formation tended to aggregate in clusters, while those with inhibition were prone to remain in a planktonic state dispersed in liquid culture.

## 3. Discussion

The number of infections caused by *Candida* species has been rising since the 1970s, probably related to the increase in susceptible patients with immunosuppressive therapies, improved diagnosis, and failure of antifungal therapies [23,24]. Generally, drugs used to treat *Candida* infections are poorly effective against biofilms, triggering the development of antifungal resistance [25]. This has led to the search for new molecules that can improve antifungal therapies through the inhibition of planktonic growth, adhesion, biofilm formation, and other virulence factors such as hyphae production by *Candida* species [26,27,28,29,30].

Essential oils represent an important source of bioactive compounds with a great variety of biological properties ranging from microbiocide activities to bacteriostatic/fungistatic, or antivirulence effects [16,17,31,32,33]. Previous studies from our lab and collaborators have demonstrated the potential use of EOs derived from Colombian plants for the control of growth and biofilm formation of pathogenic bacteria such as *E. coli*, *S. epidermidis*, and methicillin-resistant *S. aureus* (MRSA) [18,19,20].

In this study, we evaluated the effects of 50 Colombian EOs, by assessing their increased potential in planktonic growth control, biofilm formation, and inhibition of hyphal production in some *Candida* species. Many EOs reported in the literature have antifungal activity values greater than 1 mg/mL, which can be considered high depending on the cytotoxicity of the EO [34]. In this study, a concentration of 750 µg of EO/mL was used as the cutoff point for antifungal activity, since a highly active EO should exhibit an MIC_50_ lower than 375 µg/mL [35]. The *L. origanoides* EOs (LOC, LOCpT, LOT-I, LOTC, and LOT-II) were able to control fungal pathogens at concentrations from 288–750 µg/mL (Table 2). Even though these EOs were obtained from the same plant species and displayed similar antifungal activity, the EOs have different concentrations of their main secondary metabolites. This suggests that the antifungal activity was strongly related to the synergy between the different minor components, rather than to a few major components. However, EOs from *L. origanoides* thymol chemotypes (LOT-I and LOT-II) presented the highest inhibitory activity in planktonic cultures with MIC_50_ values of 141–281 µg/mL after 48 h of treatment. Higher concentrations of each bioactive EO were necessary to achieve the MFC. LOT-II presented the best fungicide activity, killing more than 99.9% of the viable cells at 563 µg/mL, probably due to the effect of thymol, which is a well-known microbiocide. *C. auris* presented lower MFCs for the EOs LOTC, LOT-I, LOT-II, and LACL than for the other two species.

Spearman’s test for antifungal activities (Appendix A) demonstrates a moderate correlation for thymol, thymyl methyl ether, carvacrol, γ-terpinene, and *p*-Cymene with general antifungal activity, consistent with the description of their antimicrobial properties in the literature [36,37]. Thymol is the main compound in LOT-I (75.3%), LOTC (49.4%), and LOT-II (71.7%), and it is probably responsible for the fungistatic and fungicide activities of these EOs. Additionally, thymol is a phenolic monoterpene widely reported in the literature with biological activities against bacteria and fungi, due to its ability to interact with cell membrane components, causing instability and integrity loss of the microbial cell [38,39,40]. Other major compounds of these bioactive EOs were *p*-cymene, carvacrol, *trans*-β-caryophyllene, and thymol methyl ether. These components have been reported as bioactive components, but have not been well characterized in terms of antifungal and antibiofilm activity against *Candida* species [39,40,41,42,43]. Further experiments are needed to identify the possible mechanisms of action of each major compound in fungal pathogens, such as *Candida* spp.

Additionally, the EOs obtained from the other species of *Lippia* presented antifungal activity against at least one *Candida* strain. It is important to highlight that antifungal activity with LACL against *C. auris* was obtained at low concentrations (MIC_50_ 188 µg/mL), but was less active against the other two *Candida* strains (MIC_50_ 750 µg/mL). This could be related to the metabolite composition of LACL, which contains a high concentration of limonene, carvone, and germacrene D. Further studies are necessary to identify the possible action mechanisms, the compound synergy, and the metabolic target that make these EOs highly active against *C. auris.* Zapata et al. also evaluated different *L. origanoides* chemotypes against clinical and reference strains of *Candida*, obtaining similar inhibition results [21].

Biofilm formation is a key attribute of the pathogenic traits of the *Candida* species which is becoming a major problem in clinical settings due to its rapid emergence and high antifungal resistance [26,27]. It has been estimated that most clinical and environmental isolates of *Candida* are related to biofilm structures [28]. Currently, the available antifungals present reduced effectivity against biofilms due to the presence of extracellular polymeric substances, high cellular density, decreased metabolism, and the presence of persistent cells [3,28]. For this reason, the search for new molecules and compounds with the capacity to prevent biofilm formation, which can be used as preventive agents or adjuvants to conventional strategies, is needed. The inoculum for biofilm formation evaluation was 100-times higher than those used for antimicrobial activity. Therefore, the inhibition percentage of planktonic cells in this assay may be lower at the same concentration.

The EOs from *L. origanoides* showed the highest inhibitory activity against all strains (MBIC_50_ 53–141 µg/mL) indicating a possible effect on the cell attachment and biofilm formation of *Candida* species. In this case, LOT-I presented a better biofilm inhibitory activity (MBIC_50_ 53 µg/mL) than LOT-II (MBIC_50_ 94–141 µg/mL). This difference could be attributed to an effect between the different metabolites of this EO that are not present in LOT-II (thymyl acetate and *trans*-β-bergamotene). Spearman’s correlation test identified that secondary metabolites, such as carvacrol, thymol, and thymyl methyl ether, had a strong correlation to biofilm inhibition; γ-Terpinene, *p*-cymene, *trans*-β-Bergamotene, *cis*-β-ocimene, humulene epoxide II, terpinen-4-ol, and carvacryl acetate presented a moderate/weak correlation to biofilm inhibition (Appendix A). These represent an opportunity for future research on biofilm formation and control in *Candida* species to determine the MICs, synergistic capacities, and action mechanisms.

The inhibition of *C. auris* biofilm formation in the presence of LOT-I and LOT-II was confirmed by SEM imaging (Figure 3). These results showed a decrease in the number of cells adhered to the surface without visible effects on the integrity of the cells. Surprisingly, the MM-II EO from *M. mollis* inhibits the adhesion and the concomitant biofilm formation exclusively on the *C. auris* strain. However, the molecular processes involved in *C. auris* biofilm formation has not yet been fully described, making it difficult to identify possible relationships between the main compounds of MM-II and biofilm formation. Secondary metabolites of MM-II EO, iso-menthone and menthone, were moderately correlated to antibiofilm activity exclusively against the *C. auris* strain. Moreover, other authors have evaluated the ability of EOs to inhibit or control the planktonic and sessile growth of *C. auris* with promising results [11,44,45,46,47]. However, most of those EOs (cinnamon, lemongrass, tea tree, cajeput, Niaouli, and White thyme) showed higher MICs (>375 µg/mL or >0.05% *v*/*v)* than those of our bioactive EOs [45,47].

*C. albicans* can form hyphae structures, which are indicators of pathogenicity and strongly related to biofilm formation [8,48]. New approaches have sought to find molecules that allow for the regulation of this virulence factor with the aim of controlling their pathogenesis [30,33,48,49]. According to the results from the inhibition of biofilm formation, EOs from *L. origanoides* inhibited hyphal formation at concentrations lower than 94 µg/mL (Figure 4). The formation of hyphal structures is closely related to the biofilm formation of *C. albicans* strains, suggesting that these EOs can be considered as possible therapeutic agents to control fungal pathogenicity. Patchouli and cinnamon EOs required at least 0.5% *v*/*v* (approximately 4500 µg/mL for our EOs) to effectively inhibit hyphae formation on *C. albicans* [33].

Non-active EOs against *Candida* strains could present other biological activities as shown for *S. aspera* (insecticide), *Piper* spp. (antioxidant and antimicrobial), *Ocimum* spp. (anti-inflammatory), *Pogostemon* spp., *H. carinosum* (antibacterial), and *Turnera* diffusa (cytotoxic activity) [18,50,51,52]. However, the biological activities of EOs derived from plants such *E. quinquenervis*, *H. dilatata*, *S. viminea*, *A. popayanensis, Simsia* spp., and *M. rhopaloides* are scarce or almost non-existent in the literature.

Compared to previous studies, biofilm inhibitory and antifungal concentrations of *L. origanoides* EOs are lower than those reported previously for bacterial strains (MIC > 375 µg/mL and MBIC > 100 µg/mL for *S. aureus*, *E. coli* and *S. epidermidis strains*) [18,19]. The results obtained in this work demonstrated the EOs’ potential to control the planktonic growth and biofilm formation of different species of *Candida*. Consequently, the EOs distilled from the different chemotypes of *L. origanoides* showed consistent biological effects against the *Candida* strains, mainly dependent on the chemical composition and concentration of the main components. Compounds like thymol, thymyl methyl ether, carvacrol, γ-terpinene, and *p*-cymene have extensive research in the literature about their antimicrobial and cytotoxic properties. However, the specific action mechanisms of their antibiofilm, antifungal, and antivirulence activity against pathogenic *Candida* strains have yet to be fully assessed. Further studies could include the evaluation of the synergy with less evaluated metabolites like iso-menthone, *trans*-β-bergamotene, *cis*-β-ocimene, humulene epoxide II, and terpinen-4-ol.

## 4. Materials and Methods

### 4.1. Plant Material

All plants used in this work were grown and harvested from experimental plots in the National Center for Agroindustialization of Aromatic and Medicinal Tropical Plant Species (CENIVAM) at the Industrial University of Santander (UIS, Bucaramanga, Colombia). Taxonomic identification was performed at the Colombian National Herbarium (National University of Colombia, UN, Bogotá, Colombia) and at the UIS Herbarium.

The *exsiccatae* and plant vouchers were deposited in the UIS Herbarium. Plants were initially collected in the countryside, and then propagated and grown in the CENIVAM experimental plots under controlled conditions. The EOs were distilled from fresh plant material. The voucher numbers, codes, plant identification, and relative compositions are shown in Appendix A.

### 4.2. Essential Oils

The EOs were distilled and characterized as previously described [18,21]. In brief, EOs were obtained by microwave-assisted hydro-distillation (MWHD) immediately after harvesting the plant material. Plants (200 g) were suspended in water (300 mL) and placed in a 2 L balloon connected to a Clevenger-type glass equipment with a Dean–Stark distillation reservoir. The plant sample was heated by microwave irradiation for 45 min (3 × 15 consecutive minutes). The EOs obtained were dried with anhydrous sodium sulfate, weighed, and stored in an amber bottle at 4 °C until further analysis. All distillations were performed in triplicate. The EOs were chemically characterized by GC/MS analysis as previously described, using a GC 6890 Plus gas chromatograph (Agilent Technologies, AT, Palo Alto, CA, USA) equipped with a mass selective detector MS 5973 Network (Palo Alto, CA, USA) using electron ionization (EI, 70 eV) [19].

For all biological tests, a stock solution (300 mg/mL) of each EO was prepared in dimethyl sulfoxide (DMSO, 100%), Sigma-Aldrich, St. Louis, MO, USA.

### 4.3. Microorganisms

The *Candida* species used in this study were *Candida albicans* ATCC 10231, *C. auris* CDC B11903, and *C. parapsilosis* ATCC 22019. These strains were stored in Sabouraud dextrose broth (SDB Sigma-Aldrich, St. Louis, MO, USA) with 20% (*v*/*v*) glycerol at −70 ± 5 °C until needed. Cells were reactivated and maintained in SDB and sabouraud dextrose agar (SDA) at 35 ± 2 °C for 48 h before each assay.

### 4.4. Determination of Antifungal Activity of EOs

Antifungal activities of the EOs were carried out by the broth microdilution method as described in the M27-A2 standard of the Clinical and Laboratory Standards Institute (CLSI) and standards of the European Committee for Antimicrobial Susceptibility Testing (EUCAST) with some modifications [53,54]. Initial screenings were performed to select EOs with antifungal activity at a concentration of 750 µg/mL prior to further evaluation. For screening, the EOs were prepared at 1500 µg/mL in sterile saline from the stock solution (300 mg/mL) dissolved in DMSO (100%; Sigma-Aldrich, St. Louis, MO, USA). After the screening, the EOs with antifungal activity were tested at 188, 281, 375, 563, and 750 µg/mL to determine their minimal inhibitory concentration (MIC_50_) and minimal fungicidal concentration (MFC) for each strain. The MIC_50_ is defined as the minimum concentration of EO being able to inhibit at least 50% of yeast growth compared to the control. The assays were performed with an initial inoculum of 0.5–5 × 10^5^ CFU/mL in 2X RPMI 1640-MOPS (Sigma-Aldrich, St. Louis, MO, USA) mixed 1:1 with 100 µL of the respective EO adjusted to achieve the concentrations described above. *Candida* without EO, with DMSO (1%; Sigma-Aldrich, St. Louis, MO, USA), and fluconazole were used as the growth control, positive, and negative, respectively. Each 96-well microtiter plate (SPL, Gyeonggi-do, Korea) with the assay was incubated at 35 ± 2 °C in the dark and read at 530 nm by a Multiskan SkyHigh Microplate Spectrophotometer (Thermo Fisher-Scientific, Waltham, MA, USA) after 0, 24, and 48 h. Experiments were carried out in triplicate. The MFC was determined by plating 10 µL of the wells without growth on SDA. After 48 h of incubation at 35 ± 2 °C, the MFC, defined as the lowest concentration of compounds that killed at least 99.9% of the original inoculum compared to the untreated control, was determined.

### 4.5. Effect of EOs on Biofilm Formation

The potential of EOs to prevent biofilm formation of *C. albicans* ATCC 10231, *C. auris* CDC B11903, and *C. parapsilosis* ATCC 22019 was evaluated in sterile 96-well round-bottom microtiter plates (SPL, Pocheon-si, Korea) [55]. The biofilm formation tests were performed using EO solutions prepared at subinhibitory concentrations (<MIC_50_) from 47 to 750 µg/mL in saline. A fresh inoculum of 5–10 × 10^6^ CFU/mL of *C. albicans* ATCC 10231, and *C. auris* CDC B11903 was prepared in 2X RPMI 1640-MOPS supplemented with glucose 2% (Merck KGaA, Darmstadt, Germany). For *C. parapsilosis* ATCC 22019, the inoculum was prepared in 2X SDB supplemented with glucose 2% at the same concentration. One hundred microliters of the respective inoculum was transferred into the plate with 100 µL of the respective EOs adjusted to achieve the concentrations described above. The plates were incubated for 24 h at 35 ± 2 °C in the dark. Cultures without EO were used as biofilm formation controls. After incubation, non-adherent cells were washed three times (3×) with sterile phosphate buffer saline (PBS pH 7.2). Biofilm formation was quantified by staining the wells with crystal violet (0.4% *w*/*v*) for 15 min. Then, the samples were washed with sterile saline to remove the excess dye, and 200 µL of 30% glacial acetic acid was added to each well [56]. Absorbance values at 590 nm were measured on a Multiskan SkyHigh Microplate Spectrophotometer (Thermo-Fisher Scientific, Waltham, MA, USA). The minimum biofilm inhibitory concentration (MBIC_50_) is defined as the minimum concentration of EO that can inhibit biofilm formation by at least 50% [18,19].

### 4.6. Scanning Electron Microscope Analysis

Scanning electron microscopy (SEM) was carried out to observe possible modifications in the *C. auris* adhesion and biofilm formation after treatment with EOs. LOT-I and LOT-II were evaluated at 94 and 144 µg/mL. Frosted glass coupons (1 cm × 2 cm × 5 mm) were deposited in 20 mL glass tubes with 2000 µL of an inoculum of 5–10 × 10^6^ CFU/mL *C. auris* CDC B11903 and 2000 µL of the EO. After 24 h of incubation at 35 ± 2 °C in the dark, the coupons were gently washed three times with a saline solution to remove planktonic cells. Samples were fixed with 2.5% glutaraldehyde for 60 min and dehydrated with an isopropanol gradient (10, 20, 30, 40, 50, 60, 70, 80, and 100%) for 10 min [18,19]. The Samples were coated with gold and observed with a scanning electron microscope FEG (field emission gun) QUANTA FEG 650 (SEM, Thermo-Fisher Scientific, Waltham, MA, USA), equipped with an Everhart Thornley ETD detector.

### 4.7. Inhibition of Hyphal Formation in C. albicans by EOs

The effect of the EO on the ability of *C. albicans* ATCC 10231 to form hyphae was determined in 96-well microtiter plates. Briefly, the cell suspension was diluted to 5–10 × 10^6^ CFU/mL in 200 µL of RPMI 1640 supplemented with 10% fetal bovine serum (FBS, Invitrogen) that contained different concentrations (47–375 µg/mL) of EO or 1% DMSO. The plate was incubated at 37 ± 2 °C with agitation (200 rpm) for 5 h and visualized using 40× lens brightfield Zeiss microscope. The photographs were taken with a Zeiss Axiocam 208 color camera [57,58].

### 4.8. Data Analysis

All the experiments were carried out in triplicate. All results were expressed as the means and their respective standard deviations for each assay. All statistical analyses were performed on Prisma, and a *p* < 0.05 was considered statistically significant. Significant changes are indicated by an asterisk in the figures. The assumption of normality and equality of variances of data were previously tested using Shapiro–Wilk and Levene test, respectively. Spearman’s correlation test was done to assess the relationships between antifungal and antibiofilm activities. The correlation coefficient can be positive (directly proportional) or negative (inversely proportional), and quantitative values suggest that the relationship between two or more variables is strong, moderate, or weak: 0.00 to 0.19 (very weak); 0.20 to 0.39 (weak); 0.40 to 0.69 (moderate); 0.70 to 0.89 (strong) and 0.90 to 1.00 (very strong) [59].

## 5. Conclusions

To summarize, this research shows that Colombian EOs present antifungal and antibiofilm properties from mainly those that contained high contents of secondary metabolites, such as carvacrol, thymol, thymyl methyl ether, γ-terpinene, and *p*-cymene. In this study, we demonstrated the potential of EOs from *L. origanoides, L. alba,* and *M. mollis* to inhibit planktonic and sessile growth of different *Candida* species. Additionally, partial inhibition of hyphal production was demonstrated. These results open the possibility for their use in the design of new therapeutic strategies against *Candida* spp.

## Figures and Tables

**Figure 1 antibiotics-12-00668-f001:**
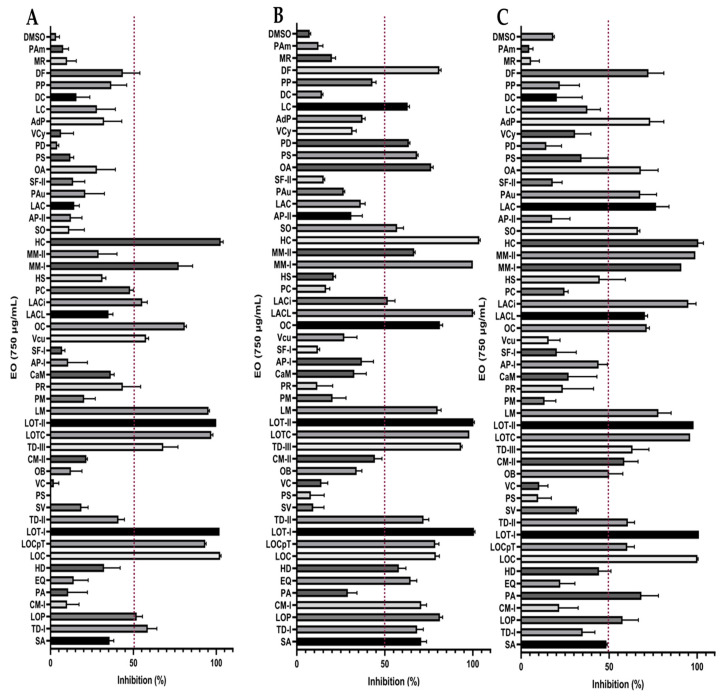
Screening of antifungal activity of 50 EOs (750 µg/mL) against (**A**) *C. albicans* ATCC 10231, (**B**) *C. parapsilosis* ATCC 22019, and (**C**) *C. auris* CDC B11903 after 48 h of treatment.

**Figure 2 antibiotics-12-00668-f002:**
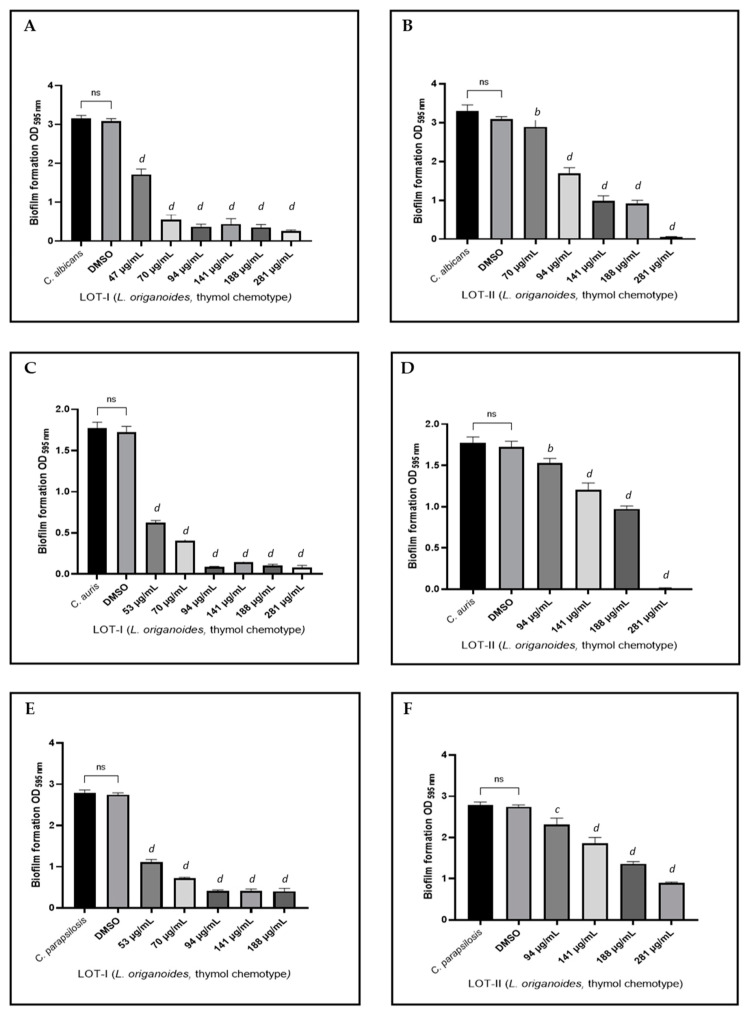
Effect of antifungal EO LOT-I (*L. origanoides*, thymol chemotype) and LOT-II (*L. origanoides*, thymol chemotype) on biofilm formation of *C. albicans* ATCC 10231 (**A**,**B**), *C. auris* CDC B11903 (**C**,**D**), and *C. parapsilosis* ATCC 22019 (**E**,**F**). Letters denote a significant difference between the treated and untreated controls, based on Tukey’s test (ns: *p* > 0.05 *b*: *p* ≤ 0.01; *c*: *p* ≤ 0.001; *d*: *p* ≤ 0.0001).

**Figure 3 antibiotics-12-00668-f003:**
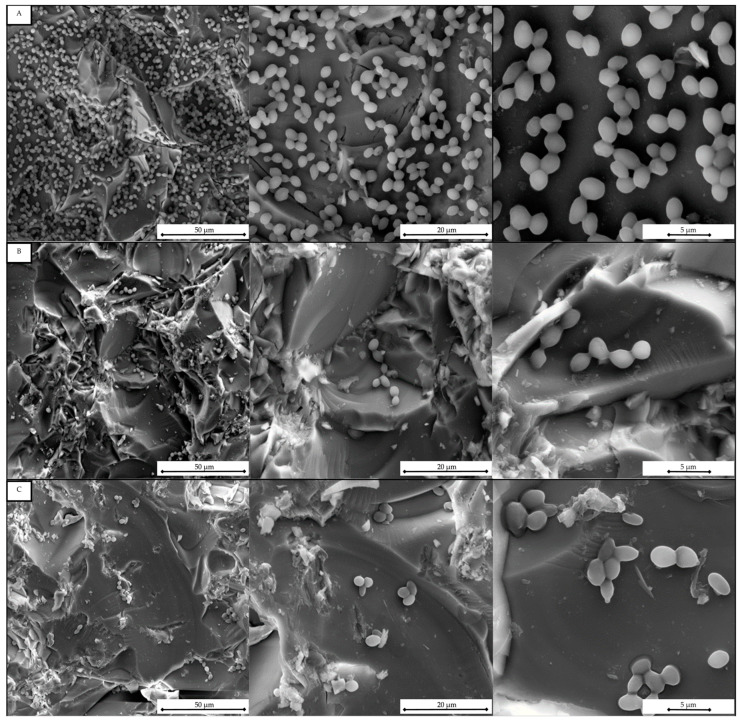
SEM images of *C. auris* biofilm (**A**) untreated (Control) (**B**) Treated with 53 µg/mL of LOT-I: and (**C**) Treated with 141 µg/mL of EO LOT-II. SEM images were recorded at 2000 and 5000, and 10,000× magnification, respectively.

**Figure 4 antibiotics-12-00668-f004:**
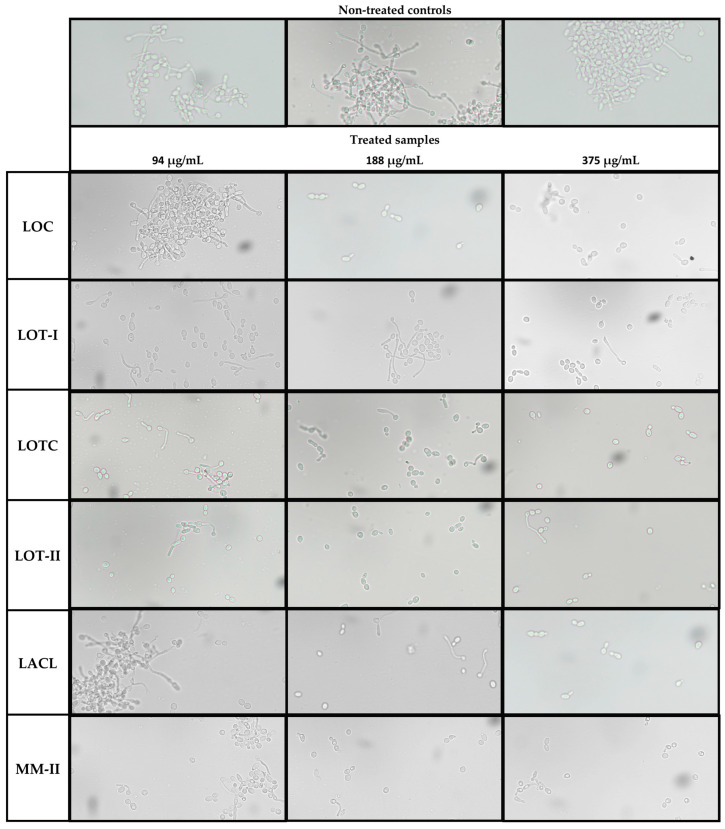
Effects of LOC, LOT-I, LOTC, LOT-II, LACL and MM-II on hyphal growth of *C. albicans* in liquid media. Yeasts were grown in RPMI 1640 medium with 10% SFB and photographed at 40× magnification after 6 h incubation.

**Table 1 antibiotics-12-00668-t001:** Information on the plants and chemical composition of EOs with biological activity (antifungal or antibiofilm) against at least one of the *Candida* strains.

Plant Code	Plant Species	Vouche Number	Major EO Compounds
TD-I	*Turnera diffusa*	UIS Herbarium 22037	Dehydrofukinone (25.4%), aristolochene (17.9%), valencene (7.4%), β-selinene (5.2%), *trans*-β-caryophyllene (4.0%), β-elemene (4.0%), premnaspirodiene (3.7%), guaiol (3.5%), germacra-4,5,10-trien-1-α-ol (3.5%), and caryophyllene oxide (3.3%).
EQ	*Elaphandra quinquenervis*	COL 587094	Germacrene D (20.7%), α-phellandrene (9.1%), α-pinene (6.8%), *trans*-β-caryophyllene (5.1%), Δ^3^-carene (4.9%), limonene (4.5%), β-cubebene (3.5%), α-humulene (2.6%), premnaspirodiene (2.6%), and δ-cadinene (2.6%).
LOC	*Lippia origanoides*, carvacrol chemotype	UIS Herbarium 22034	Carvacrol (35%), *p*-cymene (14.4%), thymol (8.0%), γ-terpinene (5.3%), *trans*-β-caryophyllene (4.4%), β-myrcene (2.4%), carvacryl acetate (2.0%), thymyl methyl ether (1.9%), and α-terpinene (1.7%).
LOCpT	*Lippia origanoides*, β-Caryophyllene-thymol chemotype	UIS Herbarium 22035	*trans*-β-Caryophyllene (15.1%), thymol (14%), 1,8-cineole (13%), *p*-cymene (12.6%), α-humulene (8.1%), α-phellandrene (7.1%), α-eudesmol (2.6%), caryophyllene oxide (2.5%), γ-terpinene (2.4%), and limonene (2.1%).
LOT-I	*Lippia origanoides*, thymol chemotype	COL 587107	Thymol (75.3%), *trans*-β-caryophyllene (5.4%), carvacrol (4.9%), α-humulene (3.2%), *p*-cymene (2.3%), thymyl acetate (1.6%), thymyl methyl ether (1.3%), caryophyllene oxide (1.3%), and *trans-*β-bergamotene (1.0%).
TD-III	*Turnera diffusa*	Herbarium UIS 22037	Aristolochene (20.6%), dehydrofukinone (17.3%), *p*-cymene (5.8%), β-selinene (5.6%), valencene (5.2%), premnaspirodiene (4.2%), caryophyllene oxide (3.6%), *trans*-β-caryophyllene (2.8%), germacra-4,5,10-trien-1-α-ol (2.4%), and α-selinene.
LOTC	*Lippia origanoides*, thymol-*p*-cymene chemotype	Herbarium UIS 22039	Thymol (49.4%), *p*-cymene (19.1%), γ-terpinene (9.2%), β-myrcene (5.2%), α-terpinene (2.9%), carvacrol (2.7%), thymyl methyl ether (1.8%), *trans*-β-caryophyllene (1.6%), *cis*-β-ocimene (1.2%), and limonene (0.9%).
LOT-II	*Lippia origanoides*, thymol chemotype	Herbarium UIS 22036	Thymol (71.7%), *p*-cymene (10.5%), carvacrol (4.4%), β-myrcene (2.1%), γ-terpinene (2.0%), caryophyllene oxide (1.6%), thymyl methyl ether (0.9%), *trans*-β-caryophyllene (0.9%), humulene epoxide II (0.7%), and terpinen-4-ol (0.7%).
LM	*Lippia micromera*	COL 560986	*p*-Cymene (26.8%), thymyl methyl ether (26.3%), thymol (17.8%), thymyl acetate (5.7%), γ-terpinene (5.4%), 1,8-cineole (5.1%), α-terpinene (2.0%), β-myrcene (2.0%), *trans*-β-caryophyllene (1.7%), α-thujene (1.3%), and caryophyllene oxide (0.9%).
OC	*Ocimum campechianum*	UIS Herbarium 20889	Eugenol (35.3%), 1,8-cineole (15.6%), β-selinene (11.0%), *trans*-β-caryophyllene (7.4%), germacrene D (5.6%), α-selinene (4.8%), β-pinene (2.4%), β-elemene (1.9%), and α-humulene (1.5%).
LACL	*Lippia alba*, carvone-limonene chemotype	UIS Herbarium 22031	Limonene (40.1%), carvone (37.7%), germacrene D (8.1%), β-bourbonene (3.0%), piperitone (1.9%), β-myrcene (0.9%), piperitenone (0.8%), linalool (0.7%), borneol (0.7%), and *trans*-β-farnesene (0.7%).
LACi	*Lippia alba*, citral chemotype	UIS Herbarium 22032	Geranial (24.5%), geraniol (19.0%), neral (11.9%), *trans*-β-caryophyllene (9.1%), germacrene D (4.3%), geranyl acetate (2.8%), α-humulene (2.8%), β-elemene (2.6%), nerol (2.5%), and limonene (2.4%).
MM-I	*Minthostachys mollis* (Benth.) Griseb.	UIS Herbarium 22041	*trans*-Piperitone oxide (49.6%), menthone (8.9%), piperitenone oxide (4.8%), *trans*-β-caryophyllene (4.0%), limonene (3.3%), thymol (2.3%), 6-hydroxycarvotanacetone (2.3%), germacrene D (2.1%), β-pinene (2.0%), linalool (1.9%), and pulegone (1.7%).
MM-II	*Minthostachys mollis* (Benth.) Griseb.	UIS Herbarium 22042	Menthone (46.1%), pulegone (13.3%), piperitone (12.1%), *trans*-β-caryophyllene (7.0%), germacrene D (3.8%), iso-menthone (3.5%), bicyclogermacrene (3.4%), α-humulene (1.9%), α-pinene (1.3%), and β-pinene (1.2%).

UIS: Industrial University of Santander (Bucaramanga, Colombia).

**Table 2 antibiotics-12-00668-t002:** MIC_50_ and MFC values for the most active EOs studied against *C. albicans* ATCC 10231, *C. parapsilosis* ATCC 22019, and *C. auris* CDC B11903 after 48 h of culture.

EO	*C. albicans* ATCC 10231		*C. parapsilosis* ATCC 22019		*C. auris* CDC B11903	
	MIC_50_ (µg/mL)	MFC (µg/mL)	MIC_50_ (µg/mL)	MFC (µg/mL)	MIC_50_ (µg/mL)	MFC (µg/mL)
EQ	NA	NA	375	NA	NA	NA
LOC	375	NA	281	NA	281	NA
LOCpT	563	750	188	NA	750	NA
LOT-I	281	750	188	750	188	563
LOTC	188	NA	281	NA	188	375
LOT-II	188	563	141	563	141	375
LM	750	NA	188	NA	750	NA
OC	563	NA	563	NA	563	NA
LACL	750	NA	750	NA	188	563
LACi	750	NA	750	NA	563	NA
MM-I	750	NA	375	750	375	NA
HC	563	NA	375	750	375	NA

NA: non-active.

**Table 3 antibiotics-12-00668-t003:** MBIC_50_ values for the most active EOs studied against *C. albicans* ATCC 10231, *C. parapsilosis* ATCC 22019, and *C. auris* CDC B11903 after 48 h of culture.

Essential Oil	*C. albicans* ATCC 10231	*C. parapsilosis* ATCC 22019	*C. auris* CDC B11903
	MBIC_50_ (µg/mL)	MBIC_50_ (µg/mL)	MBIC_50_ (µg/mL)
TD-I	NA	750	NA
LOC	281	188	NA
LOCpT	375	281	NA
LOT-I	53	53	53
CM-II	NA	750	NA
TD-III	NA	750	NA
LOTC	94	141	141
LOT-II	188	188	141
LACL	NA	750	NA
MM-II	NA	NA	188

NA: non-active.

## Data Availability

Data are contained within the article or Appendix A.

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
