# Peer review of "Antifungal and Antibiofilm Activity of Colombian Essential Oils against Different Candida Strains"

_antibiotics, 2023, doi:10.3390/antibiotics12040668_

Round 1

Reviewer 1 Report

The article is about the antifungal and antibiofilm activity of 50 essential oils from Colombia against most common Candida strains. The design of the experiment is appropiate and the manuscript reveal significant results. However, some things need major checks.

Introduction: It is weel organized and the references present are of relevance. Line 56-59 the reference for this statement is lacking.

Results: Reducing the tables with the most active essential oils makes the manuscript easier to understand . It is also useful having the complete table in the supplementary files. Into this context authors should include in the supplementary file S1 the complete composition with the compounds placed in columns or lines for all the essential oils tested, making easier to see which essential oils have similar or different compositions.

Please place the figures and tables right next to the first time they are mentioned.

In figure 2, authors should change the * method for showing significant differences to letters. In this case different letter means significative differences. It will do this easier to follow.

Discussion: This part is lacking the most of the manuscript. Authors have to perform a correlation Spearman or Pearson test to find and show the correlations between the inhibitory activities and the compounds.

Besides authors should compare with other previous research and other authors.

Authors do not discuss anything about the MFC. Please discuss it.

Are there any references about those essential oils that did not show inhibitory activities having activity against other microorganism or anythig? Why were those essential oils selected for this study?

Line 149: methicillin resistant does not go in italics.

Line 152-153: Why did not authors start with the concentration of 1 mg/mL?

Line 214: Explain more about those other researches. Are your results better than those studies?

Material and methods:

Line 261: Space is lacking "2 ºC"

Section 4.4: Authors are defining MIC50. Also define MFC.

Section 4.5: Why for the biofilm formation test were used EO at subinhibitory MIC50 concentrations?

Line 294-195: Why those conditions for C. parapsilosis

Best regards.

Author Response

Response to Reviewer 1 Comments

Point 1:  Introduction: It is weel organized and the references present are of relevance. Line 56-59 the reference for this statement is lacking.

Response 1: We agree with the referee and the reference was added.

Point 2: Results: Reducing the tables with the most active essential oils makes the manuscript easier to understand . It is also useful having the complete table in the supplementary files. Into this context authors should include in the supplementary file S1 the complete composition with the compounds placed in columns or lines for all the essential oils tested, making easier to see which essential oils have similar or different compositions.

Response 2: We agree with the referee and a new table comparing the composition of essential oils with the bioactivity was included in the Supplementary data

Point 3: Please place the figures and tables right next to the first time they are mentioned.

Response 3: The figures and tables were relocated as close as possible to the first time they are mentioned in the text.

Point 4: In figure 2, authors should change the * method for showing significant differences to letters. In this case different letter means significative differences. It will do this easier to follow.

Response 4: We agree with the referee and this figure was modified.

Point 5: Discussion: This part is lacking the most of the manuscript. Authors have to perform a correlation Spearman or Pearson test to find and show the correlations between the inhibitory activities and the compounds.

Response 5: We agree with the referee and Spearman’s test for correlation was included in Supplementary data in Tables S5 and S6; and discussed in the main text.

Point 6: Besides authors should compare with other previous research and other authors.

Response 6: We agree with the referee and additional discussion comparing our results with previous studies was included for obtained antifungal and antibiofilm activities. Lines 258-260; 249-251; 239-241

Point 7: Authors do not discuss anything about the MFC. Please discuss it.

Response 7: We agree with the referee and a discussion about MFC results was included in lines 179 - 182.

Point 8: Are there any references about those essential oils that did not show inhibitory activities having activity against other microorganism or anythig? Why were those essential oils selected for this study?

Response 8:  This research is part of a big screening of on antimicrobial activities of Colombian EOs. Essential oils from some of these plants such as Lippia spp and Pipper spp are highly characterized for their antibacterial properties. However, essential oils extracted from plants such as M. mollis , H. dilatata, S. viminea, A. popayanensis, Simsia spp. and M. rhopaloides are less published in literature.  We include some information over non-active EOs in lines 252 to 257.

Point 9: Line 149: methicillin resistant does not go in italics.

Response 9: We agree with the referee and this text was adjusted.

Point 10: Line 152-153: Why did not authors start with the concentration of 1 mg/mL?

Response 10:  Some EOs present low solubility at concentrations over 750 mg/mL, and this could lead to errors in spectrophotometric lectures, affecting reproducibility of results. Also, higher concentrations of EOs are more susceptible to volatilization and are prone to have a higher cytotoxic activity.

Point 11: Line 214: Explain more about those other researches. Are your results better than those studies?

Response 11: This part is now on line 237, and an additional explanation was added.

Point 12: Line 261: Space is lacking "2 ºC".

Response 12: We agree with the referee and the space was added.

Point 13: Section 4.4: Authors are defining MIC50. Also define MFC.

Response 13: MFC was already defined in Line 329-330.

Point 14: Section 4.5: Why for the biofilm formation test were used EO at subinhibitory MIC50 concentrations?

Response 14: To effectively identify if an EO has the capacity to inhibit biofilm formation, it is important to work with concentrations that minimally affect planktonic growth (preferably inhibition below 10%). When planktonic growth is affected, cell density and fitness are affected, therefore the inhibition would correlate more to a secondary effect of decreased viability than to a specific effect of the essential oil over adhesion or EPS production.

Point 15: Line 294-195: Why those conditions for C. parapsilosis

Response 15: Biofilm formation at these optimal conditions were previously evaluated for each strain with SDB, YPD and RPMI 1640 supplemented with glucose 2% broths. C. albicans and C. auris showed biofilm formation after 24 h with RPMI 1640 broth (crystal violet optical density > 1). C. parapsilosis presented low biofilm formation in these conditions, thus not suitable for biofilm inhibition assays. Biofilm formation for C. parapsilosis was improved with culture in SDB supplemented with glucose 2% with crystal violet (with optical density values around 2.0).  As stablished for other authors, inhibition of biofilm formation is not fully standardized yet, and optimal conditions for biofilm formation varies greatly between strains. Therefore, each assay was previously standardized for the better conditions in our lab.

Reviewer 2 Report

Part of the methods are incorporated in the results and should be moved. Especially because the material is in the end. Results and material should be demarcated. 

Author Response

Response to Reviewer 2 Comments

Point 1:  Part of the methods are incorporated in the results and should be moved. Especially because the material is in the end. Results and material should be demarcated.

Response 1: Modifications are on the new attached version of the paper.

Reviewer 3 Report

In the manuscript entitled “Antifungal and antibiofilm activity of Colombian essential oils against different Candida strains” authors studied the potential of 50 essential oils from Colombian plants against three priority fungal pathogens (C. Albicans, C.auris and C.parapsilosis) to identify the promising plants and their secondary metabolites with anti-fungal activities. The present work is of significant interest for developing anti-fungal drugs. However, I have a few major concerns that authors need to address:

  1. Table 3, entries are repeated LOCpT, LOT1, LOC etc with different values. Authors need to recheck.
  2. For SEM, on what basis the dose (94 µg/µL and 141 µg/µL) were selected? Please include scale for the images. Also include a positive control to compare. 
  3. There’s no point for Table 1 as the details are already provided in supporting file Table 1. May be the details for only the best EOs can be provided and discussed.
  4. Figure 4 images are not that clear. The cell density differs in the control (0 µg/µL). A better image quality is need for proper comparison. Also authors should have included a positive control for their experiment.

 Minor comments:

  1. In Table 2, for LOCpT MIC50 (750 µg/µL) is greater than MFC (563 µg/µL) against C.albicans. I think its an error, authors suggested to check the same.
  2. In Line. 99 MBIC50 please write the full form here or define it. It will be easier to follow as MBIC50 is used first time here.
  3. Line 211, “not yet been yet fully”, please correct the sentence.
  4. If abbreviated form is used for Essential oil (EOs), authors can avoid writing both the forms in repetition. 
  5. Typo error Line 181 “Zapata at al”
  6. Line 198; authors are suggested better not to use the term “synergistic effect” here. As the metabolites may not act well upon as per definition.
  7. The result and discussion part can be improved and proper conclusion highlighting a future direction must be added.  For example: which plant and metabolite could be further taken into consideration. Combination of which metabolites can have improved efficacy etc.

Author Response

Response to Reviewer 3 Comments

Point 1: Table 3, entries are repeated LOCpT, LOT1, LOC etc with different values. Authors need to recheck.

Response 1: We agree with the referee and the codes were rechecked and corrected.

Point 2: For SEM, on what basis the dose (94 µg/µL and 141 µg/µL) were selected? Please include scale for the images. Also include a positive control to compare.

Response 2: The concentration used for SEM images corresponds to the Minimum biofilm inhibitory concentration (MBIC50). Control for biofilm formation with DMSO is included Figure 3-A. Scale bar is presented in each picture in a more visible version.

Point 3: There’s no point for Table 1 as the details are already provided in supporting file Table 1. May be the details for only the best EOs can be provided and discussed.

Response 3: We agree with the referee and only the composition of Bioactive EOs is presented in Table 1. An improved table for the composition of active and non-active EOs is now available in Supplementary information Table S2.

Point 4:  Figure 4 images are not that clear. The cell density differs in the control (0 µg/µL). A better image quality is need for proper comparison. Also authors should have included a positive control for their experiment.

Response 4: We agree with the referee and improved the figure 4. Controls used for this experiment were Saline with 1% DMSO (figure 4. First panel).  However, there are no available compounds with standard inhibition of hyphae formation without effect on viability, so these controls were not possible for these assays.

Point 5: In Table 2, for LOCpT MIC50 (750 µg/µL) is greater than MFC (563 µg/µL) against C.albicans. I think its an error, authors suggested to check the same. 

Response 5: We agree with the referee and the data was corrected.

Point 6: In Line. 99 MBIC50 please write the full form here or define it. It will be easier to follow as MBIC50 is used first time here.  

Response 6: We agree with the referee and the full form was added.

Point 7: Line 211, “not yet been yet fully”, please correct the sentence.

Response 7: We agree with the referee and the phrase was corrected.

Point 8: If abbreviated form is used for Essential oil (EOs), authors can avoid writing both the forms in repetition.

Response 8: We agree with the referee, and this was corrected in the manuscript.

Point 9:  Typo error Line 181 “Zapata at al”

Response 9: We agree with the referee and the word was corrected.

Point 10: Line 198; authors are suggested better not to use the term “synergistic effect” here. As the metabolites may not act well upon as per definition

Response 10: We agree with the referee and the phrase was adjusted.

Point 11: The result and discussion part can be improved and proper conclusion highlighting a future direction must be added.  For example: which plant and metabolite could be further taken into consideration. Combination of which metabolites can have improved efficacy etc.

Response 11: We agree with the referee and a proper conclusion highlighting further research was added.

Round 2

Reviewer 3 Report

The revised manuscript has been improved. However, authors cited a paper in Line 171, which I find inappropriate. There are still few typos need to be corrected. Eg line 160 "microbicide". 

Author Response

Dr. Karina Yang,

Section Managing Editor

Dear Editor,

We would like to thank the Editor and each reviewer for their time and constructive comments to our manuscript. Please, find below the responses to the referee in the second check of our manuscript. All corrections were adjusted in the final version of the manuscript. 

Point 1: The revised manuscript has been improved. However, authors cited a paper in Line 171, which I find inappropriate. There are still few typos need to be corrected. Eg line 160 "microbicide".

Response 1: We agree with the referee and the paper cited in line 171 was corrected. Typos were revised corrected.

Thank you again for your time and consideration.

Kind regards,

Claudia C. Ortiz